# How Do Midwives View Their Professional Autonomy, Now and in Future?

**DOI:** 10.3390/healthcare11121800

**Published:** 2023-06-19

**Authors:** Joeri Vermeulen, Maaike Fobelets, Valerie Fleming, Ans Luyben, Lara Stas, Ronald Buyl

**Affiliations:** 1Department Health Care, Brussels Centre for Healthcare Innovation, Erasmus Brussels University of Applied Sciences and Arts, 1090 Brussels, Belgium; 2Department of Public Health, Biostatistics and Medical Informatics Research Group, Vrije Universiteit Brussel (VUB), 1090 Brussels, Belgium; 3Department of Teacher Education, Vrije Universiteit Brussel (VUB), 1040 Brussels, Belgium; 4Faculty of Health, Liverpool John Moores University, Liverpool L3 5UX, UK; 5Centre for Midwifery, Maternal & Perinatal Health, Bournemouth University, Bournemouth BH1 3LH, UK; 6Frauenzentrum (Centre for Women’s Health), Lindenhofgruppe, 3012 Bern, Switzerland; 7Support for Quantitative and Qualitative Research (SQUARE), Core Facility of the Vrije Universiteit Brussel (VUB), 1090 Brussels, Belgium

**Keywords:** midwives, midwifery, midwifery autonomy, autonomy, professionalization, maternity care, hospital-based practice, primary care, midwife-led care, sexual and reproductive health

## Abstract

Background: Internationally, midwives’ professional autonomy is being challenged, resulting in their inability to practice to their full scope of practice. This situation contrasts with the increasing international calls for strengthening the midwifery profession. The aim of this study therefore is to explore Belgian midwives’ views on their current and future autonomy. Methods: An online survey among Belgian midwives was performed. Data were collected and analyzed using a quantitative approach, while quotes from respondents were used to contextualize the quantitative data. Results: Three hundred and twelve midwives from different regions and professional settings in Belgium completed the questionnaire. Eighty-five percentage of respondents believe that they are mostly or completely autonomous. Brussels’ midwives feel the most autonomous, while those in Wallonia feel the least. Primary care midwives feel more autonomous than hospital-based midwives. Older midwives and primary care midwives feel less recognized and respected by other professionals in maternity care. The majority of our respondents believe that in future midwives should be able to work more autonomously in constructive collaboration with other professionals. Conclusion: While Belgian midwives generally rated their own professional autonomy as high, a significant majority of respondents desire more autonomy in future. In addition, our respondents want to be recognized and respected by society and other health professionals in maternity care. It is recommended to prioritize efforts in enhancing midwives’ autonomy, while also addressing the need for increased recognition and respect from society and other maternity care professionals.

## 1. Introduction

To be autonomous practitioners in maternity care, and thus a valuable resource for achieving optimum results, midwives need a strong profession worldwide [1]. A systematic literature review exploring factors influencing midwifery professionalization identified professional autonomy as a catalyst for advancing this process [2]. Autonomy grants a profession the responsibility of safeguarding the public from individuals who, for any reason, lack the requisite competence to work within the respective field [3]. Job autonomy has a positive effect on job satisfaction [4]. A positive association was found between Greek nurses’ professional autonomy and job satisfaction [5]. Furthermore, Filipino nurses with higher levels of professional autonomy were more satisfied and committed in their job than less autonomous nurses [6]. Additionally, professional autonomy positively affects employees’ subjective well-being [7]. Conversely, failure to address the needs of individuals for professional autonomy may have a negative impact on staff retention, because of job dissatisfaction [5].

A study on the professionalization of midwifery in Europe revealed concerns regarding midwives’ status and roles in practice. Additionally, their influence on healthcare systems and policymaking raised further considerations [8]. Midwives’ professional autonomy seems to be particularly limited and they face barriers in fulfilling a comprehensive role as legally defined by the European Directives on the scope of midwifery in Europe [9]. Midwifery autonomy, a cornerstone of professional practice, has been almost completely relinquished within obstetric-led care in some countries [10]. In the developed world, the medicalization of birth has been suggested as an important factor in constraining midwifery autonomy [11]. Midwives working in medicalized settings have been socialized towards increased perceptions of birth as high-risk and in need of intervention. While working within strict protocols, little room might be left for individual decision-making, and midwives working in hospital-based settings felt that they should always be able to justify their professional decisions [12]. Midwives who do not feel autonomous and experience high work pressure in an intra-partum care setting did not feel confident even when caring for women with uncomplicated births [13]. 

In Belgium, midwives’ degrees of autonomy vary; in hospital-based settings, most midwives work under the authority of the obstetrician, although this can vary across hospitals and regions [14]. The organization of health care provides a financial incentive that encourages obstetricians to undertake activities that could be carried out by midwives. Internationally, midwives in primary care settings tend to experience increased autonomy in the organization of their work [11,15,16]. In the Netherlands, midwives reported that they were more autonomous in primary care, where colleagues or obstetricians had no influence on their decisions [12]. A recent study also found that primary care midwives in Belgium felt more autonomous when compared with hospital-based working midwives [17]. In addition, primary care midwives were more satisfied with their job and work–life balance [17]. These results are in line with a literature review highlighting that midwives associated job satisfaction with autonomy, supervision and salary [18]. However, Pollard identified a range of perspectives among midwives regarding autonomous practice in the United Kingdom. The study highlighted the central issue of midwives wanting to be autonomous practitioners and, as a result, taking on increased responsibility [19]. Another study in three maternity settings in Ireland revealed that the perception amongst both midwives and obstetricians was that many midwives neither wanted to be autonomous nor take on the role of lead professional for low-risk women, because they were fearful of being accountable for decisions and implicated in adverse outcomes [20]. The authors of this study suggest that midwives’ loss of autonomy may be a self-fulfilling prophecy—i.e., midwives are resigning—and other professions will fill the gaps if the profession does not step up to the challenges it faces. Some believe that midwifery autonomy is not possible when practicing with other professionals because of historical hierarchies and power dynamics [21]. Interprofessional collaboration is only possible when the different professionals respect each other and are in equal positions. In Slovenia, for example, midwives, nurses and obstetricians are poorly aware of each other’s roles and competencies, and obstetricians do not recognize midwives as autonomous professionals for normal pregnancy, birth and postpartum [9]. A study exploring interprofessional collaboration between midwives and obstetricians in hospitals in Canada demonstrated willingness from the obstetricians to work collaboratively with midwives. The midwives, however, believed that in midwife-led settings, this collaboration would most likely restrict the power and autonomy of midwifery practice [22]. Midwives in Flanders and the Netherlands recently indicated, however, the need for a shift towards shared responsibility and autonomy [17].

In the current Belgian maternity care context, midwives might not be able to practice to the full extent of the profession’s scope of practice [14]. This situation contrasts with the growing body of evidence about the positive outcomes and cost containment of midwife-led care, and the increasing international calls for strengthening the contribution of midwifery in the public health care field [23]. Recently, a research-derived definition of midwifery autonomy in Belgium has been established, resulting in a joint understanding of the concept of midwifery autonomy. This definition of midwifery autonomy has the potential to encourage a dialog enabling to strengthen the midwifery profession in Belgium [24]. In order to explore midwives’ opinions regarding their current and future autonomy, further research in the different midwifery settings was recommended. The aim of this study therefore is to explore the current state of Belgian midwives’ autonomy and their views on their autonomy in the future.

## 2. Methods

### 2.1. Design

A descriptive observational study was conducted using an online questionnaire. Quotes from respondents were used to contextualize the quantitative data. The study is reported following the Consensus-Based Checklist for Reporting of Survey Studies (CROSS) as advocated by Sharma et al. [25], see Appendix A.

### 2.2. Participants

A non-probability convenience sampling was used. We aimed for representative participation of midwives working in the different regions of Belgium. To achieve representativeness, we included midwives working in all areas of professional practice: maternity care, reproductive medicine, gynecology and neonatology. Additionally, we targeted midwives working in primary care and those working in research and education [26].

In December 2021, a hard copy information letter containing a Quick Response (QR) code to the survey link (QualtricsXM) was sent by post to all maternity services in Belgium (n = 104, 59 in Flanders, 34 in Walloon and 11 in Brussels) [27]. Additionally, the presidents of the Flemish Organization of Midwives (1829 members in 2021), the Professional Union of Belgian Midwives (800 members in 2021) and the French-speaking Association of Catholic Midwives (105 members in 2021) were asked to distribute the call for participation to their members through newsletters and social media. Lastly, potential participants were contacted via e-mail by two researchers (JV and MF) through their personal contacts and invited to participate in the online survey between December 2021 and February 2022. The invitation included information about the study, an participant information sheet and a link to the survey site. A reminder was sent three weeks after the first invitation (January 2022).

### 2.3. Data Collection

To achieve the objectives of this study, a questionnaire was developed based on 15 items related to midwifery autonomy [24]. These items were retrieved from the literature in 2021 and translated forward and backward into Dutch and French in the context of a study on the development of a consented definition of midwifery autonomy in Belgium. An online Delphi panel of 27 content experts was engaged to evaluate the clarity and relevance of the items, leading to the validation of 15 items, following three rounds of the Delphi process. The items were related to the dimensions ‘work content’, ‘professionalism of the midwife’ and ‘relationship with others’. The work-content-related dimension of autonomy was identified as one in which the midwife is responsible, can independently take decisions and has control over her work. Likewise, the identified items related to the autonomy dimension of professionalism of the midwife were expertise, authority and competency in regard to their relationship with others. This dimension comprised items concerning the respect for the independence of midwives, their recognition and respect by other health professionals in maternity care. Together these items encompassed the essentials of midwifery autonomy as indicated by content experts, see Appendix A.

The 15 items were adapted into both Dutch and French statements by two researchers (JV and MF). The statements were externally linguistically checked (RG) and the discrepancies that had been identified were discussed and adjusted (MF, JV and RG). 

The questionnaire was delivered with the dimensions ‘work content’, ‘professionalism of the midwife’ and ‘relationship with others’ stated prior to the statements. Participants were asked to evaluate the statements on a 4-level Likert scale (1 = strongly disagree, 2 = disagree, 3 = agree and 4 = strongly agree). This scale overcame the limitations of the proportion agreement procedure, because a neutral and ambivalent midpoint was avoided [28]. When respondents indicated 1 = strongly disagree, they were invited to explain why they strongly disagreed with the statement. Respondents could freely comment on each of the 15 statements if they wished. At the end of the questionnaire, respondents were invited to rate their own autonomy and indicate to which extent they believe that a midwife in Belgium should be able to work autonomously in the nearby future. In addition, respondents could make further comments on this study. Finally, professional and sociodemographic data were gathered (gender, age, professional experience, professional setting and work region). In December 2021, prior to data collection, a pilot study was conducted involving two midwifery researchers from the Flemish Professional Association of Midwives’ scientific workgroup.

Participation in this survey was voluntary and anonymous. Consent was obtained by a cover letter explaining that completion of the questionnaire implied participants’ willingness to participate. Data were obtained in compliance with General Data Protection Regulation (European Union 2016/679). The data were stored in a secured and locked server of the Vrije Universiteit Brussel (VUB), which were only accessible to the researchers.

### 2.4. Data Analysis

To evaluate the effect of the different independent factors (age, years of professional experience, professional setting and region) on the two internally consistent and valid dimensions of autonomy, ‘work content’ and ‘professionalism of the midwife’, a stepwise model building approach for a multi-way ANOVA was used. Variables showing a significant effect using a one-way approach were considered in a full factorial (including all interactions). In the final model, all non-significant interactions were omitted. Post hoc analysis was performed using a Tukey Honestly Significant Difference (HSD) test for multiple comparisons. Prior to conducting the Tukey HSD test, Levene’s test of equality of error variances was performed, which indicated that the assumption of equal variances across groups was met (*p* > 0.05), justifying the use of the Tukey test for post hoc pairwise comparisons. The reported means fall between 5 and 20, as the sum that was calculated for each dimension consists of 5 items, and each item could be rated on a 4-level Likert scale.

Outcome variables concerning the dimension ‘relationship with others’ (items concerning the respect for the independence of midwives, their societal and professional recognition and respect by other health professionals in maternity care) were given as frequencies (percentages). We combined responses into two subgroups: the positive answers (agree and strongly agree) and negative answers (disagree and rather strongly disagree). Then, the data were analyzed by breaking it down based on age, years of professional experience, professional setting, and region. 

To test the relationship between the discrete outcomes (midwives’ views on their autonomy now and in the future) and the professional setting and region, a Chi-square test was used. The analyses were performed using SPSS Statistics for Windows, version 28.0. *p*-values ≤ 0.05 were considered significant. 

Some comments given by the participants in the survey have been used in this article to support the understanding of the research findings. Two researchers (JV and MF) selected narrative quotes from the transcripts and translated them into English.

### 2.5. Validity, Reliability and Rigor

The R programming environment (R Development Core Team 2004) was used to assess item properties, reliability and validity of the scales. First, a visual inspection of the distribution of the items revealed that some items did not follow a normal distribution. Next, the reliability of each scale’s dimension was checked using Cronbach’s Alpha coefficient (α) and Omega (based on the model using polychoric correlations). The reliability measures of both ‘work content’ and ‘professionalism of the midwife’ were considered good (work content: α = 0.80, $\omega_{u-cat}=0.81$; professionalism: α = 0.81, $\omega_{u-cat}=0.82$), while the dimension ‘relationship with others’ was suboptimal (α = 0.50, $\omega_{u-cat}=0.55$). An item inspection revealed that the items of this dimension are too heterogeneous. In the third step, the construct validity of the scale was inspected by constructing Confirmatory Factor Analyses (CFA) models using the R-package lavaan. Models using a Maximum Likelihood (ML) estimator resulted in a suboptimal model fit, due to the fact that data were non-normally distributed and assessed on only a 4-point Likert scale. Therefore, we opted to use the Diagonally Weighted Least Squares (DWLS) estimator. Different fit indices were inspected to evaluate how well the model fits the data. It was also verified if the standardized factor loadings exceed 0.50. The CFA models resulted in good model fits when the dimension ‘relationship with others’ was omitted from the model. For details on the statistical procedure, results and CFA models, see Appendix A. 

In conclusion, the scales of both dimensions ‘work content’ and ‘professionalism of the midwife’ are internally consistent, which is a measure of scale reliability. The dimension ‘relationship with others’ is not internally consistent as its items are too heterogeneous. Regarding construct validity, good model fits were obtained when the items related to the dimension ‘relationship with others’ were omitted from the model.

## 3. Results

### 3.1. Sociodemographic and Professional Characteristics of Participants

We received responses from 415 respondents, of which 312 (75.2%) were complete. Participants were all female except for three male midwives; the age group 31–40 years was the most represented (n = 96, 30.8%). Most participants had between 11 and 20 years (n = 81, 26%) of professional experience as midwives. The respondents in our study encompassed various professional backgrounds, including hospital-based midwifery (n = 185, 59.3%), primary care (n = 66, 21.2%), education (n = 45, 14.4%) and research (n = 9, 2.9%). About 10% of respondents combined different domains or were professionally practicing in a Maternal Intensive Care setting. In the hospital-based setting, the labor (n = 153, 49%) and postnatal ward (n = 148, 47.4%) were the most represented amongst participants. Nearly half of the respondents were professionally active in Flanders (n = 154, 49.3%). Of the remainder, 61 (19.7%) midwives working in the Walloon and 95 (30.4%) in the Brussels Capital region (Table 1: Sociodemographic and professional characteristics of respondents).

### 3.2. Midwives’ Views about Their Autonomy 

On the query of how midwives generally rate their own autonomy, most respondents answered that they view themselves as mostly or completely autonomous (n = 262, 84%). About 15% of respondents feel that they are mostly not (n = 45, 14.4%) or not at all autonomous (n = 5, 1.6%). Ninety-five percentage of midwives working in primary care stated that they are mostly or completely autonomous compared to midwives in hospital-based care (81.6%). This difference is statistically significant (*p* ≤ 0.001). Table 2 outlines midwives’ views about their autonomy. The older midwives become and the more professional experience they gain, the higher they rate their autonomy. Conversely, there is a decrease in the self-perceived autonomy among midwives who are aged over 50 and those with more than 21 years of experience. Midwives from the Brussels Capital region feel more autonomous (92.6%) than midwives from the Flanders (81.8%) and the Walloon regions (73.01%). One midwife highlighted the difference between workplaces:


*“After working in four different hospitals in Brussels and in the Walloon region, I think that autonomy is truly dependent on the workplace …. and depends on the leading midwives and obstetricians of the team”.*
(item 17_participant 161)

### 3.3. Midwives’ Views about Their Autonomy Related to Work Content

The autonomy items related to the content of their work involve independently making decisions, acting independently, having control over their own work, taking responsibility for their duties and being able to work according to professional regulations. For each item in this dimension, the majority of midwives scored ‘agree’ or ‘strongly agree’ (81.7–93.5%). When comparing the mean sum scores of the items related to the work content dimension, a post hoc comparison using the Tukey HSD test showed that the result for the Brussels Capital region $\bar{x}$ = 16.27 was significantly different from Flanders $\bar{x}$ = 15.56 (*p* = 0.04) and the Walloon $\bar{x}$ = 14.69 (*p* < 0.01) regions. The mean score for primary care midwives was significantly different from midwives from both hospital-based and primary care ($\bar{x}$ = 17.31 vs. $\bar{x}$ = 15.41, *p* ≤ 0.01), hospital-based midwives ($\bar{x}$ = 17.31 vs. $\bar{x}$ = 15.18, *p* ≤ 0.01) and midwives from other professional settings, e.g., education and research ($\bar{x}$ = 17.31 vs. $\bar{x}$ = 14.89, *p* ≤ 0.01), which was also pointed out by one of the participants:


*“In the hospital I work under the supervision of the obstetrician, which limits my ability to take independent decisions. Approval from the obstetrician is always required. In primary care I work independently according to the guidelines …, I can also prescribe medication, … primary and hospital care vary day and night”.*
(item 1_participant 273)

No statistically significant difference was found in relation to age (*p* = 0.58) and years of professional experience of respondents (*p* = 0.07) regarding midwives’ perception about their autonomy related to work content. Table 3 outlines midwives’ views about their autonomy related to work content and the professionalism of the midwife.

### 3.4. Midwives’ Views about Their Autonomy Related to the Professionalism of the Midwife

The autonomy items related to the professionalism of the midwife include taking responsibility and being responsible for own decisions, having expertise, feeling competent and having authority. For each item of this dimension, the majority of respondents agreed or strongly agreed with the statements (88.1–96.8%). When comparing the mean sum scores of the items related to the professionalism of the midwife dimension, a post hoc comparison using the Tukey HSD test showed that the result for the Brussels Capital region $\bar{x}$ = 16.85 was significantly different from Walloon $\bar{x}$ = 15.77 (*p* < 0.01). Two respondents expressed this as follows:


*“Specifically at X (Brussels hospital): no hierarchy between doctors and midwives, but complementarity +++ and teamwork”.*
(item 11_participant 177)


*“As the advocate of physiology an active collaboration with the team of doctors in X (Brussels hospital) is pursued”.*
(item 9_participant 11)

Midwives working in Flanders scored higher on this autonomy dimension than midwives working in Walloon, but the difference was not statistically significant (*p* = 0.06). The mean scores of primary care midwives were significantly higher compared to hospital-based midwives ($\bar{x}$ = 17.23 vs. $\bar{x}$ = 16.24, *p* ≤ 0.01) and those from other professional settings ($\bar{x}$ = 17.23 vs. $\bar{x}$ = 15.97, *p* = 0.02). The mean score of midwives with a professional experience of fewer than 5 years scored significantly less on this dimension than for midwives with more professional experience ($\bar{x}$ = 15.23 vs. more than $\bar{x}$ = 16.60 for the other age categories, *p* ≤ 0.01). A midwife with less than 5 years of professional experience pointed this out:


*“I still lack a bit of experience. This is a job we learn every day. The more experience we acquire, the more comfortable we will be in the job”.*
(item 8_participant 319)

### 3.5. Midwives’ Views about Their Autonomy in Relationship with Others

The dimension of autonomy related to the relationship with others involves not being supervised, being socially and professionally recognized and being respected by other health professionals and a professional association of midwives which defines the professional rules. The older midwives are and the more professional experience they have, the more they feel recognized by society and professionally, see Table 4. Nevertheless, a decline in this self-perceived societal and professional recognition is observed after more than 30 years of professional experience, which was highlighted by one of the midwives:


*“We are often supervised by assistants who respond very medically and do not consider our experience or expertise, which often leads to frustration …”.*
(item 11_participant 297)

Primary care midwives feel less supervised by doctors or other health professionals (84.4%) than hospital-based midwives (31.8%). Considerably fewer midwives from primary care feel recognized by society (50.0% vs. 63.7% hospital-based midwives) and professionally (57.8% vs. 68.6% hospital-based midwives). Several midwives reflected on their feelings in this regard in their comments:


*“Despite my proven competencies, I am not rewarded by the government, which is inexcusable. No extra fee for me as an accredited lactation consultant, no additional fee if you have a Master’s degree, …”.*
(item 9_participant 174)


*“Not every woman or doctor accepts the expertise of a midwife. I think this is the most difficult thing in my profession, the daily struggle to prove that what we do is responsible, safe and qualitative care”.*
(item 10_participant 2)


*“I often get comments like ‘Is a midwife allowed to perform a childbirth?’ ‘Isn’t that dangerous?’ or ‘I would prefer to give birth with an obstetrician anyway’ …”.*
(item 12_participant 9)

Additionally, primary care midwives feel less respected by other health professionals in maternity care (54.7% vs. 82.9% hospital-based midwives); this lack of respect was expressed by several primary care midwives:


*“When one does not know primary care, many health professionals are suspicious about my professional functioning, this is due to a lack of information and understanding of primary care”.*
(item 13_participant 287)


*“You are not always considered as an authority in maternity care, rather as someone with an alternative, not evidence based vision”.*
(item 10_participant 286)

Midwives working in the Walloon feel considerably less recognized by society (39.4% vs. 69.8% midwives from Flanders) and professionally (29.5% vs. 81.4% midwives from Flanders) and less respected by other health professionals in maternity care (65.6% vs. 86.3% of midwives from Brussels Capital). Midwives working in Flanders, among other regions, feel the most recognized by society and professionally, while midwives from Brussels feel most respected by other health professionals in maternity care.


*“Respect and trust are not self-evident, but are built up by good and constructive cooperation”.*
(item 14_participant 122)

Primary care midwives are less in agreement with the statement that a professional association of midwives defines the rules governing the exercise of their profession, and they pointed this out:


*“There is little support from the professional organization towards primary care midwives, the organization is almost exclusively governed by hospital-based midwives”.*
(item 15_participant 71)

The majority of Walloon midwives (80.3%) agree that a professional association of midwives defines the rules governing the exercise of their profession compared to about 70% (69.9%) in Flanders. Some compared this situation in Belgium with what they knew from other countries:


*“We need a professional organization such as in the United Kingdom (RCM [Royal College of Midwives]) or the Netherlands (KNOV, [Royal Dutch Organization of Midwives]), where most staff is professionally involved in policy, vision, research, ….”.*
(item 15_participant 174)


*“There should be mandatory membership, as in France, so that they [professional association] have more means to defend and develop our profession”.*
(item 15_participant 129)

### 3.6. Midwives’ Views about Their Autonomy in the Future

Most midwives feel that a midwife should be able to work completely autonomously (n = 156, 50%) or mostly autonomously (n = 148, 47.4%) in the future. Primary care midwives are significantly more convinced than hospital-based midwives and those who combine primary and hospital-based care (*p* ≤ 0.01) that midwives in the future should be able to work more autonomously: 


*“I believe that midwifery autonomy can be improved, such as midwifery led care units or midwife-led care, where we can take autonomous decisions, of course in the event of a low risk pregnancy/childbirth. However, this also requires a different view on the financing of maternity care”.*
(item 16_participant 22)

Many respondents highlighted that the call for more professional autonomy does not exclude collaboration with other health professionals in maternity care:


*“For me autonomy means ‘on my own responsibility, without supervision of a doctor’, but that does not mean that there should be no good cooperation with other disciplines”.*
(item 9_participant 122)


*“Good cooperation and agreements with other health professionals disciplines does not exclude autonomy”.*
(item 17_participant 122)

## 4. Discussion

This is the first study to explore the current state of Belgian midwives’ autonomy and their views on their autonomy in the future. Belgian midwives are cohesive regarding their self-perceived autonomy, with about 85% of respondents stating that they are mostly or completely autonomous. Primary care midwives feel more autonomous than midwives from hospital-based care and those combining both domains. These findings are echoed in international literature, where commonly primary care midwives experience more professional autonomy than hospital-based midwives [12,15,16]. Primary care midwives in particular are accustomed to decide about their practice and how to organize their work. Autonomy is embedded in their daily work, e.g., for primary care midwives in New Zealand professional autonomy is seen as self-evident [15]. As in the Netherlands, primary care respondents felt significantly less supervised by doctors or other health professionals than hospital-based midwives [12]. Most Belgian primary care midwives work in a model of care whereby maternity care is provided by the same midwife [10], which leads to experiencing higher levels of autonomy [29].

The degree to which hospital-based midwives are autonomous is variable and depends on the extent of authority given to them by their place of practice [15]. While hospital-based midwives feel less autonomous than primary care midwives, still over 80% rate their autonomy as high. Even if an autonomous practice is more difficult for hospital-based midwives, autonomy still is apparent and considered important for them [15]. In the Netherlands, lower professional autonomy rates were found in hospital-based midwives compared to primary care midwives, but 70.5% of hospital-based midwives felt autonomous though [30]. These high rates might illustrate that professional autonomy is important for health professionals as autonomy is related to the sense of belonging [31], job satisfaction [32] and professional well-being [33]. Internationally, mixed views are reported among midwives about whether they practice autonomously or not, based on their working environment and interpretation of the autonomy concept [19]. Conversely, it has been argued that midwives who never observed primary maternity care may not truly understand midwifery autonomy [20].

As previously outlined, Belgium midwives’ degree of autonomy may vary across regions [14]. Midwives working in the Brussels Capital region in general perceive themselves as more autonomous than midwives from the Flemish and Walloon regions. The context of a healthcare setting might be sociocultural and politically influenced which may impact midwives’ autonomy [34]. Further research is warranted to determine how the sociocultural profile of the Brussels Capital region may affect midwives’ work and autonomy.

Midwives with a professional experience of fewer than 5 years scored significantly less on the professionalism dimension than midwives with more professional experience. This is not surprising as the items in this dimension are related to expertise, competence and authority. It is to be expected that one will acquire these professional competencies over time. The older midwives become and the more professional experience they gain, the higher they rate their own autonomy. Older midwives and midwives with more professional experience feel more social and professional recognition. There is evidence that shows that more experience led to higher levels of job satisfaction in Dutch midwives [35]. The decline in our participants’ self-perceived societal and professional recognition after more than 30 years of professional experience is thus surprising and unexplained. Conversely, a decrease in the self-perceived autonomy of midwives older than 50 years and those with more than 21 years of experience was equally noticed. There might be systematic differences in how midwives perceive their work at different stages of their life course [36]. A scoping review aimed at identifying challenges faced by older nurses and midwives in the workplace identified, in addition to physical and promotional difficulties, a lack of respect from other health professionals and management. The lack of respect is mainly related to a lack of acceptance and recognition for their years of practice and expertise [37]. 

In contrast, it is concerning that primary care midwives feel less societally and professionally recognized and also experience a lack of respect from other health professionals in the field of maternity care. Primary care midwives in the Netherlands emphasized that satisfaction with collaboration with other health professionals in maternity care varies within regions and echelons of care [38]. The question of why midwives working in Walloon do not feel equally autonomous, recognized and respected as midwives from other regions remains unanswered. All midwives in Belgium work within the same legislative framework, and the organization of hospital-based settings is similar. Further research is required to identify the factors that contribute to the disparities in autonomy, recognition and respect among midwives in different regions of Belgium. Regarding societal and professional recognition, midwives feel that their competencies are ignored by society. While the lack of adequate compensation for Belgian midwives is acknowledged as hindering midwifery professionalization [39], women’s limited knowledge about midwives’ competencies may affect midwives’ recognition and autonomy [34]. The fact that women in Brussels do not consider midwives to play a central role in uncomplicated pregnancy prefer obstetricians for uncomplicated labor may explain why they are doubtful about midwives’ competencies [40]. 

Most hospital-based midwives and midwives from research and education believe that a professional association of midwives defines the rules governing the exercise of their profession. There is less agreement on this among primary care midwives. However, there is a call for a more professional functioning of the Belgian professional associations of midwives. The WHO emphasizes the importance of a strong midwifery association defending midwives’ rights, making their voices heard in order to promote professional autonomy and self-regulation of midwives [1]. Midwifery associations need to be strong political actors to promote and negotiate the position of all their members. Interconnection between midwifery organizations, other health professionals in maternity care and policymakers with women’s groups is recommended [40].

Feeling professionally respected by other health professionals in maternity care is important for our respondents. Likewise, in Greece, the strongest effect on high job satisfaction among midwives was noted to be the respect they received from women and other health professionals [32]. Midwives work often with other health professionals in practice who have a different ideology that can be challenging if professionals are required to work according to ideologies different from those of their own profession [3]. Professional relationships between similar professions play an important role in the professionalization of occupation [41]. It has been acknowledged that when different professional groups know each other, this results in mutual respect and constructive collaboration. Interprofessional collaboration should be further explored since midwives’ autonomy and participatory decision-making may be challenged which affect their professionalism [2]. A Swedish study exploring midwives’ professional role and identity revealed that midwives’ role in childbirth care had gradually decreased in favor of obstetricians [41]. Nevertheless, midwives found that the dialog with them was gradually improved, which led to better teamwork and joint decisions.

The majority of our respondents believe that in future, midwives should be able to work completely or mostly autonomously. This is more expressed by primary care midwives. In the literature, opposing views are reported, as to whether midwives want to practice autonomously and take on more responsibility [19]. Other studies identified reluctance for increased autonomy, mostly because of fear of being held responsible for professional choices [10,12]. Such a lack of clarity and fears are not echoed in our results. While a majority of Belgian midwives perceive themselves as autonomous, they desire even more autonomy in future in constructive collaboration with other health professionals in maternity care. A WHO report confirms indeed that midwives worldwide want to be more autonomous, which means that they would require professional respect to be afforded by obstetricians [1]. 

As Walloon midwives, older midwives and primary care midwives feel less recognized and not respected by other health professionals, research is warranted to explore this blind spot. We recommend exploring the views of stakeholders (e.g., general practitioners, obstetricians, pediatricians, policy advisors, hospital managers and women’s groups) about midwifery and midwifery autonomy. Specific attention should be paid to older midwives, primary care midwives and midwives working in Walloon.

### Limitations

This is the first study to explore how Belgian midwives perceive their professional autonomy. Nevertheless, attention should be drawn to the limits of our study. We employed a quantitative approach for data collection and analysis, supplementing it with respondent quotes. The combination of quantitative and qualitative approaches offers strengths, such as complementary data, triangulation of findings, depth in understanding and contextualization of quantitative data. However, this approach may pose challenges in integration and interpretation, risk of bias in selecting and interpreting quotes and limited generalizability of qualitative findings [42]. Therefore, a systematic in-depth analysis of the qualitative data from the comments through thematic analysis is planned for the near future. 

From the 435 answers received, only 312 (75.2%) completed the whole questionnaire. It is somewhat surprising that nearly 25% of respondents did not complete the survey. It had been demonstrated that the inclusion of a QR code will make respondents more likely to participate with a smartphone. Conversely, breakdown rates for respondents participating by smartphone are considerably higher compared with other devices [43]. Despite this breakdown, an acceptable variation sampling was achieved. Respondents represented each of the identified midwifery domains, namely, hospital-based midwifery, primary care, education and research. The outstanding participation of primary care midwives participated in this study, 21% vs. 9% of the national average midwives working in primary care, could be explained because professional autonomy is most significant in their job satisfaction [35]. Indeed, it has been reported that Belgian primary care midwives are more satisfied with their job than hospital-based midwives [17]. A potential limitation of this study is that we not considered the participants’ country of origin and country of education, which may have implications for the generalizability of the findings on midwives’ views regarding their professional autonomy.

The dimension ‘relationship with others’ was not internally consistent as the items were too heterogeneous. To identify if midwives feel supervised, recognized and respected and if a midwifery association defines the professional rules, descriptive analyses were used (frequencies and percentages). To support future use of the professional relationships measure of autonomy, additionally, qualitative methods should be considered to understand how midwives view their independence, recognition and respect by other maternity care professionals.

## 5. Conclusions

While Belgian midwives generally rate their own professional autonomy as high, we observed significant differences between hospital-based and primary care midwives and the region where she works. Older midwives and midwives working in the Walloon Region feel less societally and professionally recognized than others in the sample. While primary care midwives feel significantly more autonomous than hospital-based midwives, they feel less societally and professionally recognized and less respected by other health professionals in maternity care. A significant majority of respondents desire more autonomy in future and want to be respected by society and other health professionals in maternity care.

Societal and professional recognition emerged as vital factors irrespective of midwives’ age, region or professional setting. Professional organizations’ role in the ongoing process of midwifery professionalization is key to promoting and negotiating midwives’ position. Additionally, the views of stakeholders in maternity care, including policymakers and women, about the midwifery profession need to be explored to obtain a full understanding of the complex phenomenon of midwifery autonomy.

## Figures and Tables

**Table 1 healthcare-11-01800-t001:** Sociodemographic and professional characteristics of respondents who completed the questionnaire.

		n (%) 312 (100)
Gender	Female	309 (99.0)
	Male	3 (1.0)
Age (years)	20–30	74 (23.7)
	31–40	96 (30.8)
	41–50	65 (20.8)
	>50	77 (24.6)
Professional experience as a midwife (years)	<5	71 (22.7)
	5–10	60 (19.2)
	11–20	81 (26.0)
	21–30	47 (15.1)
	>30	53 (17.0)
Professional setting ^1^	Hospital-based care	185 (59.3)
	Primary care	66 (21.2)
	Both hospital-based and primary care	28 (9.0)
	Education and/or research	55 (17.3)
	Other	32 (10.3)
Professional activities in hospital-based care ^2^	Postnatal ward	148 (47.4)
	Labor ward	153 (49.0)
	Antenatal consultation	61 (19.6)
	Ultrasound	7 (2.2)
	Reproductive medicine	2 (0.6)
	Gynecology	13 (4.2)
	Neonatology	62 (19.9)
	Other	36 (11.5)
Region	Brussels Capital	95 (30.4)
	Flanders	156 (49.9)
	Walloon	61 (19.7)

^1^ The total is >100% as a midwife can be employed in several settings. ^2^ The total is >100% as a midwife can be professionally active in different hospital-based settings.

**Table 2 healthcare-11-01800-t002:** Midwives’ views about their autonomy now and in the future.

		How Do You Generally Rate Your Autonomy as a Midwife?	To What Extent Do You Think That a Midwife Should Be Able to Work Autonomously in Belgium?
		n (%) ^1^	n (%) ^1^
Age (years)	20–30	59 (79.7)	72 (97.3)
	31–40	80 (83.3)	95 (99.0)
	41–50	58 (89.2)	64 (98.5)
	>50	65 (84.4)	73 (94.8)
Professional experience as a midwife (years)	<5	54 (76.1)	70 (98.6)
	5–10	50 (83.3)	59 (98.3)
	11–20	72 (90.0)	79 (97.5)
	21–30	43 (83.7)	46 (97.9)
	>30	43 (81.1)	50 (94.3)
Professional setting	Hospital-based care	151 (81.6)	180 (97.3)
	Primary care	63 (95.5)	66 (100)
	Both hospital-based and primary care	24 (85.7)	26 (92.9)
	Other	72 (83.7)	84 (97.7)
Region	Brussels Capital	88 (92.6)	91 (95.8)
	Flanders	126 (81.8)	152 (98.7)
	Walloon	46 (73.0)	59 (96.7)

^1^ Sum of respondents who answered ‘mostly autonomous’ or ‘completely autonomous’ on the statements.

**Table 3 healthcare-11-01800-t003:** Midwives’ views about their autonomy related to work content and the professionalism of the midwife.

Dimension: Work Content		Work Content Score (5–20)			
		Mean	Standard Deviation	N	*p*-Value
Age (years)	20–30	15.28	2.71	74	0.58
	31–40	15.71	2.60	96	
	41–50	15.97	2.46	65	
	>50	15.47	2.25	77	
Professional experience as a midwife (years)	<5	14.89	2.87	71	0.07
	5–10	15.88	2.42	60	
	11–20	15.99	2.47	81	
	21–30	15.51	2.23	47	
	>30	15.75	2.32	53	
Professional setting	Hospital-based care	15.18	2.45	182	<0.01 *
	Primary care	17.31	2.06	64	
	Both hospital-based and primary care	15.41	2.87	29	
	Other	14.89	2.01	37	
Region	Flanders	15.56	2.50	156	<0.01 *
	Brussels Capital	16.27	2.22	95	
	Walloon	14.69	2.71	61	
Dimension: professionalism of the midwife		Professionalism of the midwife score (5–20)			
		Mean	Standard Deviation	N	*p*-value
Age (years)	20–30	15.93	2.34	74	0.16
	31–40	16.43	2.27	96	
	41–50	16.86	2.14	65	
	>50	16.69	2.17	77	
Professional experience as a midwife (years)	<5	15.23	2.14	71	<0.01 *
	5–10	16.63	2.46	60	
	11–20	16.94	1.95	81	
	21–30	16.89	2.16	47	
	>30	16.83	2.13	53	
Professional setting	Hospital-based care	16.24	2.16	182	<0.01 *
	Primary care	17.23	2.66	64	
	Both hospital-based and primary care	16.79	1.80	29	
	Other	15.97	1.92	37	
Region	Flanders	16.50	2.29	156	0.03 *
	Brussels Capital	16.85	2.13	95	
	Walloon	15.77	2.20	61	

* = Statistically significant: *p* ≤ 0.05 (Tukey HSD).

**Table 4 healthcare-11-01800-t004:** Midwives’ views about their autonomy in relationship with others.

		I Am Not Supervised by Doctors or Other Health Professionals n (%) ^1^	I Am Recognized by Society n (%) ^1^	I Am Professionally Recognized n (%) ^1^	Other Health Professionals in Maternity Care Respect the Role of the Midwife n (%) ^1^	A Legitimately Established Professional Association of Midwives Defines the Rules Governing the Exercise of Their Profession. This in Consultation with the Competent Authorities n (%) ^1^
Age (years)	20–30	29 (39.2)	40 (54.1)	49 (66.2)	55 (74.3)	58 (78.4)
	31–40	44 (45.8)	56 (58.3)	58 (60.4)	76 (79.2)	65 (67.7)
	41–50	33 (50.7)	40 (61.5)	44 (67.7)	50 (76.9)	49 (75.4)
	>50	35 (45.5)	54 (70.1)	58 (75.3)	58 (75.3)	51 (66.2)
Professional experience as a midwife (years)	<5	26 (37.7)	39 (56.5)	44 (62.0)	54 (76.1)	53 (74.6)
	5–10	30 (50.0)	30 (50.0)	41 (68.3)	46 (66.7)	45 (75.0)
	11–20	38 (46.9)	47 (58.0)	49 (60.5)	64 (79.0)	49 (60.5)
	21–30	22 (46.8)	38 (80.9)	38/ (80.9)	36 (76.6)	31 (66.0)
	>30	25 (47.2)	36 (67.9)	37 (69.8)	39 (73.6)	36 (67.9)
Professional setting	Hospital-based care	58 (31.8)	116 (63.7)	125 (68.6)	151 (82.9)	133 (73.0)
	Primary care	254 (84.4)	32 (50.0)	37 (57.8)	35 (54.7)	42 (65.6)
	Both hospital-based and primary care	12 (41.4)	18 (62.0)	21 (72.4)	23 (79.3)	20 (68.9)
	Other	141 (45.2)	24 (64.9)	26 (70.3)	30 (81.1)	28 (75.8)
Region	Brussels Capital	41 (44.2)	57 (60.0)	64 (67.3)	82 (86.3)	65 (68.4)
	Flanders	69 (44.2)	109 (69.8)	127 (81.4)	117 (75)	109 (69.9)
	Walloon	30 (49.2)	24 (39.4)	18 (29.5)	40 (65.6)	49 (80.3)

^1^ Sum of respondents who ‘agreed’ or ‘strongly agreed’ with the statements.

## Data Availability

The data presented in this study are available on request from the corresponding author.

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
