# Peer review of "How Do Midwives View Their Professional Autonomy, Now and in Future?"

_healthcare, 2023, doi:10.3390/healthcare11121800_

Round 1

Reviewer 1 Report

The present manuscript reports interesting data for the rader. However, the legnth of the manuscript is chaotic. Major revisins and reforming are required. First of all, please reform the abstract following a structured form. 

Introduction section must be shorter including a maximum of 4 paragraphs. Many information of introduction section represents a full-lenght discussion.

In methodology please mention more details about the sampling methods and if a pilot study was conducted.

Results section has also to be shorten. Table 1 is not necassary and can be transfered in Supplementary materials. Moreover I am not sure about the scientific value of the mentioned answers that were reported word-by-word.

Similarly with the other sections, discussion section has to be shorten.

Author Response

Dear reviewer,

The authors wish to thank you sincerely for the time you have taken to review our paper, as well as for your feedback and suggestions for improvement. We took all feedback and suggestions into consideration, and we believe that the manuscript has improved significantly (track changes in the manuscript). Additionally, the manuscript was linguistically and critically proofread by a native English speaker (Prof. Dr. Valerie Fleming, Liverpool John Moores University, Liverpool, United Kingdom).

Below you can find a point-by-point response to the reviewer’s comments. We hope the changes we have made are in line with your expectations.

Comment 1: The present manuscript reports interesting data for the reader. However, the length of the manuscript is chaotic. Major revisions and reforming are required. First of all, please reform the abstract following a structured form.

Authors’ comments: Thank you for your suggestion. We now included: background, methods, results and conclusion to structure the abstract.

Page: 1

Comment 2: Introduction section must be shorter including a maximum of 4 paragraphs. Many information of introduction section represents a full-length discussion.

Authors’ comments: Thank you for this suggestion, we removed 2 paragraphs which were very detailed and limited the introduction now to 4 paragraphs as suggested.

Page: 2

Comment 3: In methodology please mention more details about the sampling methods and if a pilot study was conducted.

Authors’ comments: Thank you for this suggestion, we now included that we used a non-probability convenience sampling. Additionally, we included that: ‘In December 2021, prior to data collection, a pilot study was conducted involving two midwifery researchers from the Flemish Professional Association of Midwives' scientific workgroup’.

Page: 3 and 5

Comment 4: Results section has also to be shorten. Table 1 is not necessary and can be transferred in Supplementary materials. Moreover I am not sure about the scientific value of the mentioned answers that were reported word-by-word.

Authors’ comments: Thank you for your suggestions. We transferred table 1 in Supplementary material (Document S2: Questionnaire) and we shortened the Results section.

We additionally elaborated on the use of qualitative data in the ‘Discussion’ section: ‘We employed a quantitative approach for data collection and analysis, supplementing it with respondent quotes. The combination of quantitative and qualitative approaches offers strengths such as complementary data, triangulation of findings, depth in understanding and contextualization of quantitative data. However, this approach may pose challenges in integration and interpretation, risk of bias in selecting and interpreting quotes and limited generalizability for qualitative findings [1]’.

Page: Supplementary material (Document S2: Questionnaire)

Page: 6-15

Page: 15

Comment 5: Similarly with the other sections, discussion section has to be shorten.

Authors’ comments: Thank you for your suggestion. We have taken it into consideration and made the necessary adjustments by appropriately shortening the discussion section.

Page: 13-15

Thank you for all your comments and suggestions, we hope we have answered all of them reliably and as you expected.

References

  1. McEvoy P, Richards D. A critical realist rationale for using a combination of quantitative and qualitative methods. Journal of Research in Nursing 2006;11(1):66-78. doi: 10.1177/1744987106060192.

Reviewer 2 Report

I would like to thank the esteemed authors for their meticulous and self-sacrificing work. I enjoyed reading the article immensely. It can be published as it is, but there are 3 references to the same author in the references section (source numbers 8, 10 and 30) and this may attract the attention of the readers. If it is rearranged by giving 1 citation instead of 3 separate citations from the authors in the sources (vermaulen et al.), it will receive less. 

Author Response

Dear reviewer,

The authors wish to thank you sincerely for the time you have taken to review our paper, as well as for your feedback and suggestions for improvement. We took all feedback and suggestions into consideration, and we believe that the manuscript has improved significantly (track changes in the manuscript). Additionally, the manuscript was linguistically and critically proofread by a native English speaker (Prof. Dr. Valerie Fleming, Liverpool John Moores University, Liverpool, United Kingdom).

Below you can find a point-by-point response to the reviewer’s comments. We hope the changes we have made are in line with your expectations.

Comment 1: I would like to thank the esteemed authors for their meticulous and self-sacrificing work. I enjoyed reading the article immensely. It can be published as it is, but there are 3 references to the same author in the references section (source numbers 8, 10 and 30) and this may attract the attention of the readers. If it is rearranged by giving 1 citation instead of 3 separate citations from the authors in the sources (Vermeulen et al.), it will receive less.

Authors’ comments: Thank you for your kind words and appreciated suggestion. We now indeed removed references 8 and 10 to avoid multiple self-citation. Reference 30 remains appropriate in the specific Belgian context.

Page: 3

Thank you for all your comments and suggestions, we hope we have answered all of them reliably and as you expected.

Reviewer 3 Report

Review:

The manuscript is systematic and clear written about a still important subject in the practice of midwifery; autonomy is one of the core elements of a midwife-professional. It was a pleasure to read, and to review.

I have a few minor comments to probably improve the paper:

-       I suggest to add midwife-led care to the keywords;

-       Line 52: nuance the sentence, e.g. midwives’ professional autonomy seems to be particularly limited etc? I think that there are differences in countries;

-       Line 62-64: ….who do not feel not autonomous….it seems one ‘not’ too many, or the sentence is not fully clear;

-       the methods section is well constructed, it was a pleasure to read;

-       line 194: did the authors consider including participants’ country of origin and country of education in the professional demographic data?

-       Line 539-541: A Swedish study exploring mid- 539 wives’ professional role and identity over the last 20–25 year revealed that midwives’ role 540 in childbirth care had decreased in favor of obstetricians [49].

The ‘last 20-25 year’ seems contradicting with the year of the research and publication of reference 49 (e-pub 2007). I suggest to adjust this sentence.

Author Response

Dear reviewer,

The authors wish to thank you sincerely for the time you have taken to review our paper, as well as for your feedback and suggestions for improvement. We took all feedback and suggestions into consideration, and we believe that the manuscript has improved significantly (track changes in the manuscript). Additionally, the manuscript was linguistically and critically proofread by a native English speaker (Prof. Dr. Valerie Fleming, Liverpool John Moores University, Liverpool, United Kingdom).

Below you can find a point-by-point response to the reviewer’s comments. We hope the changes we have made are in line with your expectations.

Comment 1: The manuscript is systematic and clear written about a still important subject in the practice of midwifery; autonomy is one of the core elements of a midwife-professional. It was a pleasure to read, and to review.

Authors’ comments: Thank you for your kind words!

Comment 2: I have a few minor comments to probably improve the paper:

I suggest to add midwife-led care to the keywords;

Line 52: nuance the sentence, e.g. midwives’ professional autonomy seems to be particularly limited etc? I think that there are differences in countries;

Line 62-64: ….who do not feel not autonomous….it seems one ‘not’ too many, or the sentence is not fully clear;

Authors’ comments: Thank for your advice and suggestions, which we happily included.

We included midwife-led care to the keywords, we nuanced the sentence as suggested and indeed in line 62-64 there was one ‘not’ too much, which we removed.

Page: Abstract and page 2

Comment 3: The methods section is well constructed, it was a pleasure to read. Line 194: did the authors consider including participants’ country of origin and country of education in the professional demographic data?

Authors’ comments: This is an interesting comment, thank you for mentioning. No, we did not considered participants country of origin and country of education, which indeed would be interesting for the interpreting of the results. We consider this a limitation of our research and now included this issue in the limits section. ‘A potential limitation of this study is that we not considered the participants' country of origin and country of education, which may have implications for the generalizability of the findings on midwives' views regarding their professional autonomy’.

Page: 15-16

Comment 4: Line 539-541: A Swedish study exploring mid- 539 wives’ professional role and identity over the last 20–25 year revealed that midwives’ role 540 in childbirth care had decreased in favor of obstetricians [49].

The ‘last 20-25 year’ seems contradicting with the year of the research and publication of reference 49 (e-pub 2007). I suggest to adjust this sentence.

Authors’ comments: Thank you for pointing out this contradiction, we adjusted the sentence. ‘A Swedish study exploring midwives’ professional role and identity revealed that midwives’ role in childbirth care had gradually decreased in favor of obstetricians’.

Page:15

Thank you for all your comments and suggestions, we hope we have answered all of them reliably and as you expected.

Reviewer 4 Report

Thank you for the opportunity to review this interesting research.

line 18 ... 'challenged resulting in them not being able to practice to their full scope' .. or something like that

line 23 'quant approach with quotes from responses being used to conceptualize this data'. Should be Three hundred and twelve ... or change the sentence around. Same comment for next sentence

line 25 maybe start with specifically or something else to make it clear you are breaking the data down to regions.

line 27 remove one of 'midwives' from this sentence

include a recommendation in abstract and maybe bit of discussion

line 42 - put in a lick at start of sentence, same line 43 as these sentences read like dot points and not read well. Same comment next sentence

line 50 paragraph is more then two sentences

line 62 'midwives who do not feel autonomous and experience high work pressure in an intrauterine care setting, did not feel confident even when caring for women with uncomplicated births

para from line 65 - how do the majority of midwives practice - assume in a hospital in medical models of care and not midwifery models. Is primary health midwifery model of care.

line 78 - what are the implications here of international people - otherwise why add this statement

line 84 does nt make sense - is the impact increased workload, and am not sure how this relates to primary care - does this mean that more care is transferred to whoever it is that provides the care in the community because women go home earlier

line 85 sentence not clear. Does not follow the previous sentence so needs a link

line 86 should this sentence be in the previous para

line 92 - what is the implication here

line 94 para may be best to be earlier when first discuss how maternity care is provided

line 98 maybe clear to say in midwifery models of care as that is specifically what the literature states

line 99 start with 'for instance' maybe

line 106 not clear .. is this Polish midwives in UK or is this an outside reflection of the Polish midwives towards UK.

line 128 needs a why

line 142 incomplete sentence. Should be questionnaire (consistency of language here needed). Assume from this there were open ended questions. Needs more of a description of this process and what was in this questionnaire, how it was developed, tested and so forth

line 159 unethical to contact potential participants through personal contacts

line 162 should be participant information sheet, informed consent not needed as the consent is implied

line 177 should refer to the table and outline what it contains

line 196 should be consent was obtained

assume also included some demographics

line 234 ? new para

be consistent with use of 0 before decimal point or not

line 241 ? new para

line 262 remove first 'professional'

line 272 refer to table and describe

line 282 do not start sentence with a number - needs writing out or rearrange sentence.

line 284 - new sentence from 'this'

line 289 maybe add in the reason for this

refer to table and describe

line 300 definition of autonomy may have been useful to have put in in the introduction to provide context

line 303 'dimension the majority of midwives'

line 321 refer to table and describe. Consistency of 0 before decimal point

line 335 you actually have two comments here from two different people - so not a response

line 393 was there any indication as to why this might be the case

refer to table

line 425 end of bracket after 50%

line 447 'with about'

line 450 needs a word before 'commonly'

line 489 socially and professionally

line 501 ? in contrast'

line 509 ? is unclear - not sure what is meant by open here

need recommendations

I wish you well with your ongoing research

Author Response

Dear reviewer,

The authors wish to thank you sincerely for the time you have taken to review our paper, as well as for your feedback and suggestions for improvement. We took all feedback and suggestions into consideration, and we believe that the manuscript has improved significantly (track changes in the manuscript).

The manuscript was linguistically and critically proofread by a native English speaker (Prof. Dr. Valerie Fleming, Liverpool John Moores University, Liverpool, United Kingdom).

We would additionally like to highlight the main differences between our previous study [1] and present study.

Overall, the first study broadens the scope by including the views of various stakeholders in maternity care, while the present study system focuses on midwives' individual perspectives on autonomy. The methodologies employed, qualitative focus group interviews versus quantitative survey, contribute to different data collection and analysis approaches in each study. The first study provides insights from stakeholders regarding aspects such as education, competence, experience, safe care, and collaboration. The present study highlights regional and professional setting differences in perceived autonomy, as well as concerns about recognition and respect.

Below you can find a point-by-point response to the reviewer’s comments. We hope the changes we have made are in line with your expectations.

line 18 ... 'challenged resulting in them not being able to practice to their full scope' .. or something like that

Authors’ comments: Thank you for this suggestion, we amended the sentence

Page: Abstract

line 23 'quant approach with quotes from responses being used to conceptualize this data'. Should be Three hundred and twelve ... or change the sentence around. Same comment for next sentence

Authors’ comments: Thank you for this suggestion, we amended both the sentences as suggested

Page: Abstract

line 25 maybe start with specifically or something else to make it clear you are breaking the data down to regions.

Authors’ comments: Thank you for this suggestion, we now included ‘and regions’ to make it clear we  will break the data down to regions.

Page: Abstract

Line 27 remove one of 'midwives' from this sentence

Authors’ comments: Thank you for this suggestion, we amended the sentence and removed one of the ‘midwives’ as suggested

Page: Abstract

include a recommendation in abstract and maybe bit of discussion

Authors’ comments: Thank you for this suggestion, we included a one sentence recommendation in the abstract

Page: Abstract

line 42 - put in a lick at start of sentence, same line 43 as these sentences read like dot points and not read well. Same comment next sentence

Authors’ comments: Thank you for this suggestion, we amended the sentence

Page: 1

line 50 paragraph is more than two sentences

Authors’ comments: Thank you for this suggestion, we made two shorter sentences out of it.

Page: 2

line 62 'midwives who do not feel autonomous and experience high work pressure in an intrauterine care setting, did not feel confident even when caring for women with uncomplicated births

Authors’ comments: Thank you for this suggestion, we amended the sentence

Page: 2

para from line 65 - how do the majority of midwives practice - assume in a hospital in medical models of care and not midwifery models. Is primary health midwifery model of care.

Authors’ comments: Thank you for your comment, we removed the paragraph

Page: 2

line 78 - what are the implications here of international people - otherwise why add this statement

Authors’ comments: Thank you for your comment, we removed the paragraph

Page: 2

line 84 does nt make sense - is the impact increased workload, and am not sure how this relates to primary care - does this mean that more care is transferred to whoever it is that provides the care in the community because women go home earlier

Authors’ comments: Thank you for your comment, we removed the paragraph

Page: 2

line 85 sentence not clear. Does not follow the previous sentence so needs a link

Authors’ comments: Thank you for your comment, we removed the paragraph

Page: 2

line 86 should this sentence be in the previous para

Authors’ comments: Thank you for your comment, we removed the paragraph

Page: 2

line 92 - what is the implication here

Authors’ comments: Thank you for your comment, we removed the paragraph

Page: 2

line 94 para may be best to be earlier when first discuss how maternity care is provided

Authors’ comments: Thank you for your comment, we removed the paragraph

Page: 2

line 98 maybe clear to say in midwifery models of care as that is specifically what the literature states

Authors’ comments: Thank you for your comment, we removed the paragraph

Page: 2

line 99 start with 'for instance' maybe

Authors’ comments: Thank you for your comment, we removed the paragraph

Page: 2

line 106 not clear .. is this Polish midwives in UK or is this an outside reflection of the Polish midwives towards UK.

Authors’ comments: Thank you for your comment, we did not meant ‘Polish midwives’, but are referring to Pollard’s publication [2]. For clarity, we rephrased the sentence, tough

Page: 3

line 128 needs a why

Authors’ comments: Thank you for your comment, we now elaborated on this statement

Page: 3

line 142 incomplete sentence. Should be questionnaire (consistency of language here needed). Assume from this there were open ended questions. Needs more of a description of this process and what was in this questionnaire, how it was developed, tested and so forth

Authors’ comments: Thank you for pointing this out, we amended the sentence. We elaborated now more on the questionnaire in data collection (description of the process) and we included additionally the testing of the questionnaire

Page: 4-5

line 159 unethical to contact potential participants through personal contacts

Authors’ comments: Thank you for pointing this out. Our study (and recruiting strategy) was approved by the Ethics Committee of the University Hospital Brussels / Vrije Universiteit Brussel (VUB), Belgium on November 24th 2021 (registration number: B.U.N. 143/202/100/0490). All participants were provided with clear and unbiased information about the study, had the option to voluntarily participate, and their personal information was handled confidentially. Participation in this online questionnaire was voluntary and anonymous. As such, the researchers were unable to determine who participated. Additionally, due to the high participation rate, the identification of individual participants would not be possible.

At all times the principles of informed consent, privacy, anonymity and confidentiality were ensured.

line 162 should be participant information sheet, informed consent not needed as the consent is implied

Authors’ comments: Thank you for your comment, we made the change as suggested (participation information sheet)

Page: 4

line 177 should refer to the table and outline what it contains

Authors’ comments: Thank you for your comment, however we moved the table to supplementary material

Page: Supplementary material

line 196 should be consent was obtained

Authors’ comments: Thank you for your comment, we made the change as suggested (consent was obtained)

Page: 5

line 234 ? new para

Authors’ comments: Thank you for your comment, this paragraph is now restructured

Page: 6

be consistent with use of 0 before decimal point or not

Authors’ comments: Thank you for your comment, we checked this all over the manuscript

line 241 ? new para

Authors’ comments: Thank you for your comment, this paragraph is now restructured

Page: 6

line 262 remove first 'professional'

Authors’ comments: Thank you for your comment, we made the change and removed the first professional

Page: 5

line 272 refer to table and describe

Authors’ comments: Thank you for your comment, we included this now

Page: 7

line 282 do not start sentence with a number - needs writing out or rearrange sentence.

Authors’ comments: Thank you for your comment, we made the necessary change

Page: 7

line 284 - new sentence from 'this'

Authors’ comments: Thank you for your comment, we shortened the sentence

Page: 7

line 289 maybe add in the reason for this

Authors’ comments: Thank you for your comment, we elaborated more on this in the discussion section

Page: 14-15

refer to table and describe

Authors’ comments: Thank you for your comment, we included this now

Page: 7

line 300 definition of autonomy may have been useful to have put in in the introduction to provide context

Authors’ comments: Thank you for this insightful reflection. We chose to provide a comprehensive description and context of the dimensions of the definition since it serves as the foundation for our questionnaire (Data collection). Unfortunately, we cannot include the exact definition as published in the Journal of Advanced Nursing due to copyright restrictions [3]. We appreciate your understanding

Page: 7

line 303 'dimension the majority of midwives'

Authors’ comments: Thank you for your comment, we included this now (the)

Page: 7

line 321 refer to table and describe. Consistency of 0 before decimal point

Authors’ comments: Thank you for your comment, we included this now and checked all over the manuscript

Page: 9

line 335 you actually have two comments here from two different people - so not a response

Authors’ comments: Thank you for your comment, we made the change making it plural (two respondents)

Page: 10

line 393 was there any indication as to why this might be the case

Authors’ comments: Thank you for your comment, we elaborated more on this in the discussion section

Page: 14-15

refer to table

Authors’ comments: Thank you for your comment, we now referred to the table

Page: 10

line 425 end of bracket after 50%

Authors’ comments: Thank you for your suggestion, we included this now

Page: 13

line 447 'with about'

Authors’ comments: Thank you for your suggestion, we included this now (with)

Page: 13

line 450 needs a word before 'commonly'

Authors’ comments: Thank you for your suggestion, we included this now (where)

Page: 13

line 489 socially and professionally

Authors’ comments: Thank you for your suggestion, we included this now (socially and professionally)

Page: 14

line 501 ? in contrast'

Authors’ comments: Thank you for your suggestion, we included this now (in contrast)

Page: 14

line 509 ? is unclear - not sure what is meant by open here

Authors’ comments: Thank you for your comment, we rephrased this now (remains unanswered)

Page: 14

Need recommendations

Authors’ comments: Thank you for highlighting the need for a recommendation here. We now recommend that: ‘further research is required to identify the factors that contribute to the disparities in autonomy, recognition, and respect among midwives in different regions of Belgium’.

I wish you well with your ongoing research

Authors’ comments: Thank you so much for your much appreciated suggestions and support!

Thank you for all your comments and suggestions, we hope we have answered all of them reliably and as you expected.

References

  1. Vermeulen J, Buyl R, Luyben A, Fleming V, Fobelets M. Key Maternity Care Stakeholders' Views on Midwives' Professional Autonomy. Healthcare 2023;11(9)doi: 10.3390/healthcare11091231.
  2. Pollard K. Searching for autonomy. Midwifery 2003;19(2):113-24. doi: 10.1016/s0266-6138(02)00103-1.
  3. Vermeulen J, Buyl R, Luyben A, Fleming V, Fobelets M. Defining midwifery autonomy in Belgium: Consensus of a modified Delphi study. Journal of Advanced Nursing 2022:1-12. doi: 10.1111/jan.15209.

Reviewer 5 Report

Dear Authors 

Thank you for allowing me to review this manuscript. It has been a pleasure for me to read this paper, which deals with an important and relevant topic for midwifery. I would like to make a few comments with the aim of improving this manuscript.

-The introduction is well presented and focuses well on the topic under study. The references are correct. There are several self-citations, but their use in my opinion is justified, as the authors are referents in their field in their country (they also cite an important reference study for European midwives).  

-Methods: In my opinion, the design of this study is adequate and in accordance with the objectives set out. I have been impressed by the effort made in the process of Validity and reliability. I have to say that there are certain aspects of the validation process and the factor analysis with which I do not agree or which perhaps require explanation, but that in no way impedes the publication of the manuscript. I will comment on them separately in case they are useful to the authors for further studies. For me this is not a validation study; it is a study that has used a new tool and where the authors try to ensure as far as possible the validity and reliability of the tool used, which is to be welcomed. 

 Some aspects that the authors can introduce:

-In participants they can indicate the type of sampling used (non-probability convenience sampling). The data they provide on the total number of midwives likely to participate are very relevant, but in results they should report the participation rate based on this. It will be a very low participation rate, but this does not prevent this study from being published.

-The generation of the items could be explained in a little more depth. I understand that it is based on the results of an already published Delphi study. A brief explanation would improve this manuscript. If you consider it, this could be applied to the generation of the dimensions.

-Please indicate which variables were collected in addition to the questionnaire items. I think this would help in the subsequent understanding of the results.

- Data analysis is adequate. The quantitative variables (age, Professional experience as a midwife ) should have been collected in raw values, not with intervals (categorical variable), for greater analytical capacity, but this is just my opinion. Indicate the normality test performed which advised the use of the Tukey test.

- Validity, reliability and rigour: If possible, please indicate if any test was performed to check the sample adequacy of the factor analysis (e.g. Kaiser Meyer Olkin test or Barlett's statistic). Since you do not present data in the manuscript on the CFA (it is supplementary material), I do not consider it necessary to indicate the fit indices used in the manuscript.

Results: are well written and well understood. Please indicate in the footnote of table 4 which statistical test was used in this case.  Perhaps indicating in each verbatim the item-participant number in the following way would improve comprehension Example: (item 1_participant 273)

Discussion: Adequate and the main aspects to be considered are discussed. Limitations appear. 

Factor analysis considerations (for future studies): The sample size is perhaps low for a Factorial analisys. This will determine the use of a linear or non-linear model. You should indicate what type of correlation matrix was used (polychoric or Pearson's), for the type of data presented with high asymmetry, it is appropriate to use polychoric correlations. Also the extraction system of the factors and the rotation used should be indicated. Crombach is a reliability coefficient that is no longer in use, you should try to complement it with omega or if you do advanced factor analysis you can calculate ORION coefficients. I leave you a reference in case you are interested:

González-de la Torre H, Hernández-Rodríguez MI, Moreno-Canino AM, Portela-Lomba AM, Berenguer-Pérez M, Verdú-Soriano J. Cross-Cultural Adaptation and Validation of the Perceptions of Empowerment in Midwifery Scale in the Spanish Context (PEMS-e). Healthcare (Basel). 2023 May 18;11(10):1464. doi: 10.3390/healthcare11101464.

I congratulate you on your work. I hope that my comments will help to improve an already very good research. Best regards

Author Response

Dear reviewer,

The authors wish to thank you sincerely for the time you have taken to review our paper, as well as for your feedback and suggestions for improvement. We took all feedback and suggestions into consideration, and we believe that the manuscript has improved significantly (track changes in the manuscript). Additionally, the manuscript was linguistically and critically proofread by a native English speaker (Prof. Dr. Valerie Fleming, Liverpool John Moores University, Liverpool, United Kingdom).

Below you can find a point-by-point response to the reviewer’s comments. We hope the changes we have made are in line with your expectations.

Comment 1: Dear Authors , Thank you for allowing me to review this manuscript. It has been a pleasure for me to read this paper, which deals with an important and relevant topic for midwifery. I would like to make a few comments with the aim of improving this manuscript.

The introduction is well presented and focuses well on the topic under study. The references are correct. There are several self-citations, but their use in my opinion is justified, as the authors are referents in their field in their country (they also cite an important reference study for European midwives).

Authors’ comments: Thank you for your kind comments.

Comment 2: Methods: In my opinion, the design of this study is adequate and in accordance with the objectives set out. I have been impressed by the effort made in the process of Validity and reliability. I have to say that there are certain aspects of the validation process and the factor analysis with which I do not agree or which perhaps require explanation, but that in no way impedes the publication of the manuscript. I will comment on them separately in case they are useful to the authors for further studies. For me this is not a validation study; it is a study that has used a new tool and where the authors try to ensure as far as possible the validity and reliability of the tool used, which is to be welcomed.

 Some aspects that the authors can introduce:

-In participants they can indicate the type of sampling used (non-probability convenience sampling). The data they provide on the total number of midwives likely to participate are very relevant, but in results they should report the participation rate based on this. It will be a very low participation rate, but this does not prevent this study from being published.

-The generation of the items could be explained in a little more depth. I understand that it is based on the results of an already published Delphi study. A brief explanation would improve this manuscript. If you consider it, this could be applied to the generation of the dimensions.

-Please indicate which variables were collected in addition to the questionnaire items. I think this would help in the subsequent understanding of the results.

Authors’ comments: Thank you for your kind words and appreciated suggestions!

We now included that we used a non-probability convenience sampling. Additionally, we elaborated on the generation of the items: ‘An online Delphi panel of 27 content experts was engaged to evaluate the clarity and relevance of the items, leading to the validation of 15 items following three rounds of the Delphi process’.

We now specified that we additionally gathered data regarding, gender, age, professional experience, professional setting and work region.

Page: 3, 4 and 5

Comment 3: Data analysis is adequate. The quantitative variables (age, Professional experience as a midwife ) should have been collected in raw values, not with intervals (categorical variable), for greater analytical capacity, but this is just my opinion. Indicate the normality test performed which advised the use of the Tukey test.

Validity, reliability and rigour: If possible, please indicate if any test was performed to check the sample adequacy of the factor analysis (e.g. Kaiser Meyer Olkin test or Barlett's statistic). Since you do not present data in the manuscript on the CFA (it is supplementary material), I do not consider it necessary to indicate the fit indices used in the manuscript.

Authors’ comments: Authors’ comments: Thank you for your valued opinion! The use of the Tukey test was advised by a Levene’s test of Equality of Error Variances. We now elaborated on this rationale: ‘Prior to conducting the Tukey HSD test, a Levene's test of equality of error variances was performed, which indicated that the assumption of equal variances across groups was met (p > 0.05), justifying the use of the Tukey test for post-hoc pairwise comparisons’.

Thank you for your suggestion on the Kaiser Meyer Olkin test or Barlett's statistic. These tests can provide valuable information when conducting exploratory factor analysis (EFA) to determine whether data are suitable for conducting such an analysis. In our study, however, we used a confirmatory factor analysis (CFA) because we have specific hypothesis about what factors are underlying the used measures. To assess the adequacy of the measurement model using a CFA, the model fit should be inspected using different model fit indices. These fit indices allow us to evaluate how well the proposed model fits the data. First, if the chi-square is not significant, the model fit is considered acceptable since the observed covariance matrix is considered similar to the model-implied covariance matrix. For the Comparative Fit Index (CFI) it is advised that the CFI exceeds .90 or, ideally, .95. Next, a value of the Tucker Lewis Index (TLI) between .90 and .95 is considered as a marginal fit, values exceeding .95 represent a good fit (Kenny, 2014). Concerning the Root Mean-Square Error of Approximation (RMSEA) a value below 0.04 describes a good fit and below 0.08 a moderate fit (Kline, 2010).

Both test and results are included in Supplementary file 3 ‘Document S3 Reliability and construct validity of the autonomy questionnaire’.

References

Bentler, P. M. (1989). EQS structural equations program manual. BMDP Statistical Software.

Kenny, D. A. (2014). Measuring Model Fit. http://davidakenny.net/cm/fit.htm

Kline, R. B. (2010). Principles and Practice of Structural Equation Modeling. The Guilford Press.

We additionally removed the section related to the fit indices.

Page:5 and Supplementary file 3 ‘Document S3 Reliability and construct validity of the autonomy questionnaire’.

Comment 4: Results: are well written and well understood. Please indicate in the footnote of table 4 which statistical test was used in this case. Perhaps indicating in each verbatim the item-participant number in the following way would improve comprehension Example: (item 1_participant 273)

Authors’ comments: Thank you for these suggestions. We included in table 4 (now table 3) that the Tukey HSD test was performed. We also indicated in each verbatim the number of the item and the participant as suggested.

Page: 9-10

Comment 5: Discussion: Adequate and the main aspects to be considered are discussed. Limitations appear.

Factor analysis considerations (for future studies): The sample size is perhaps low for a Factorial analysis. This will determine the use of a linear or non-linear model. You should indicate what type of correlation matrix was used (polychoric or Pearson's), for the type of data presented with high asymmetry, it is appropriate to use polychoric correlations. Also the extraction system of the factors and the rotation used should be indicated.

Cronbach is a reliability coefficient that is no longer in use, you should try to complement it with omega or if you do advanced factor analysis you can calculate ORION coefficients. I leave you a reference in case you are interested:

González-de la Torre H, Hernández-Rodríguez MI, Moreno-Canino AM, Portela-Lomba AM, Berenguer-Pérez M, Verdú-Soriano J. Cross-Cultural Adaptation and Validation of the Perceptions of Empowerment in Midwifery Scale in the Spanish Context (PEMS-e). Healthcare (Basel). 2023 May 18;11(10):1464. doi: 10.3390/healthcare11101464.

Authors’ comments: A covariance matrix was used in the factor analysis, which measures the covariance between variables, which captures both the linear relationship and the variability between variables. Using a covariance matrix assumes that the variables are measured on the same scale and have similar variances. This is included in Supplementary file 3 ‘Document S3 Reliability and construct validity of the autonomy questionnaire’.

Thank you for the reference, which give valuable insight! Indeed, we opted to use the Cronbach’s alfa.as this measure is still accepted as a reliability coefficient. It remains one of the most commonly employed measures to assess the internal consistency of a scale or test

We believe there might be some confusion with an exploratory factor analyses. Using this kind of analysis, the extraction system and rotation indeed are crucial information. However, we performed a confirmatory factor analysis. As described in the manuscript, we use a diagonally weighted least squares estimation technique, which is a robust Weighted Least Squares (WLS) estimation. This technique uses simpler matrix calculations than does the full WLS method. More specific, robust WLS methods only uses the diagonal in the weight matrix from full WLS estimation and generate corrected standard errors and model test statistics (see also the book of Kline, 2016, chapter Estimation and Local Fit Testing). Such methods have generally performed well in computer simulation studies except when the sample size is only about N = 200 (Muthén, du Toit, & Spisic, 1997; Finney & DiStefano, 2013). Therefore, we believe that for such an analysis, our sample size was adequate.

References:

Finney, S. J., & DiStefano, C. (2013). Nonnormal and categorical data in structural equation modeling. In G. R. Hancock & R. O. Mueller (Eds.), Structural equation modeling: A second course (2nd ed., pp. 439–492). Charlotte, NC: IAP.

Kline, R. B. (2016). Principles and practice of structural equation modeling (4th ed.). Guilford Press

Muthén, B. O., du Toit, S. H. C., & Spisic, D. (1997). Robust inference using weighted least squares and quadratic estimating equations in latent variable modeling with categorical and continuous outcomes. Retrieved from www.statmodel.com/bmuthen/articles/Article_075.pdf

Thank you for the reference, which give valuable insight! Indeed, we opted to use the Cronbach’s alfa.as this is one of the most commonly used measures of internal consistency[. However, we acknowledge that this measure is not the most optimal one to report. Therefore, we now also calculated and included the omega measure in our text.

Page: 5-6

Comment 6: I congratulate you on your work. I hope that my comments will help to improve an already very good research. Best regards

Authors’ comments: Thank you for your valuable and welcomed suggestion! Best wishes.

Thank you for all your comments and suggestions, we hope we have answered all of them reliably and as you expected.

Round 2

Reviewer 1 Report

Authros made an effort to improve their manuscript accordingly.